# Coronin1C Is a GDP-Specific Rab44 Effector That Controls Osteoclast Formation by Regulating Cell Motility in Macrophages

**DOI:** 10.3390/ijms23126619

**Published:** 2022-06-14

**Authors:** Yu Yamaguchi, Tomoko Kadowaki, Nozomi Aibara, Kaname Ohyama, Kuniaki Okamoto, Eiko Sakai, Takayuki Tsukuba

**Affiliations:** 1Department of Dental Pharmacology, Graduate School of Biomedical Sciences, Nagasaki University, Nagasaki 852-8588, Japan; yu-y@nagasaki-u.ac.jp (Y.Y.); eiko-s@nagasaki-u.ac.jp (E.S.); 2Department of Frontier Oral Science, Graduate School of Biomedical Sciences, Nagasaki University, Nagasaki 852-8588, Japan; tomokok@nagasaki-u.ac.jp; 3Department of Pharmacy Practice, Graduate School of Biomedical Sciences, Nagasaki University, Nagasaki 852-8588, Japan; nozomi-ai@nagasaki-u.ac.jp (N.A.); k-ohyama@nagasaki-u.ac.jp (K.O.); 4Department of Dental Pharmacology, Graduate School of Medicine, Dentistry and Pharmaceutical Sciences, Okayama University, 2-5-1 Shikata-cho, Kita-ku, Okayama 700-8525, Japan; k-oka@okayama-u.ac.jp

**Keywords:** Rab44, Coronin1C, osteoclast differentiation, effector, cell motility

## Abstract

Osteoclasts are multinucleated bone-resorbing cells that are formed by the fusion of macrophages. Recently, we identified *Rab44*, a large Rab GTPase, as an upregulated gene during osteoclast differentiation that negatively regulates osteoclast differentiation. However, the molecular mechanisms by which Rab44 negatively regulates osteoclast differentiation remain unknown. Here, we found that the GDP form of Rab44 interacted with the actin-binding protein, Coronin1C, in murine macrophages. Immunoprecipitation experiments revealed that the interaction of Rab44 and Coronin1C occurred in wild-type and a dominant-negative (DN) mutant of Rab44, but not in a constitutively active (CA) mutant of Rab44. Consistent with these findings, the expression of the CA mutant inhibited osteoclast differentiation, whereas that of the DN mutant enhanced this differentiation. Using a phase-contrast microscope, Coronin1C-knockdown osteoclasts apparently impaired multinuclear formation. Moreover, Coronin1C knockdown impaired the migration and chemotaxis of RAW-D macrophages. An in vivo experimental system demonstrated that Coronin1C knockdown suppresses osteoclastogenesis. Therefore, the decreased cell formation and fusion of Coronin1C-depleted osteoclasts might be due to the decreased migration of Coronin1C-knockdown macrophages. These results indicate that Coronin1C is a GDP-specific Rab44 effector that controls osteoclast formation by regulating cell motility in macrophages.

## 1. Introduction

Osteoclasts are multinucleated bone-resorbing cells derived from hematopoietic precursors of the monocyte-macrophage lineage [1]. Osteoclast differentiation from macrophages is mainly regulated by receptor activator of nuclear factor κ-B ligand (RANKL) [2]. During bone resorption, osteoclasts form resorption lacunae, which are surrounded by the ruffled border membrane that is specific equipment for an acidic extracellular microenvironment [3,4]. Therefore, the lysosome biogenesis of osteoclasts is essential for osteoclastogenesis.

Recently, our research group revealed that some Rab GTPases in osteoclasts are involved in osteoclast differentiation via membrane transport of important factors. Rab11A and Rab11B in osteoclasts have a similar function that negatively regulates osteoclastogenesis through proteolytic control of the cell surface receptors, such as RANK and c-fms, despite exhibiting differential localization [5,6]. Rab11A localizes in early and late endosomes, whereas Rab11B is localized in early and late endosomes, the Golgi complex, and the endoplasmic reticulum [5,6,7]. Rab27A deficiency induces multinucleation of osteoclasts but impairs resorption activity via the abnormal membrane transport of lysosomal proteins and cell surface receptors [8]. Rab44 was recently identified as an upregulated gene during osteoclast differentiation that negatively regulates osteoclast differentiation by modulating intracellular calcium levels [9]. Unlike small GTPases, including Rab1-43, Rab44 is a large Rab GTPase containing multiple domains, such as the EF-hand domain, coiled-coil domain, and Rab-GTPase domain [10]. However, little information is available on the molecular mechanisms by which Rab44 regulates osteoclast differentiation.

Coronins are a family of actin-binding proteins that control cell motility and actin-dependent processes, such as Arp2/3 interactions and the actin-depolymerizing factor (ADF)/cofilin pathway [11]. Coronin1C (also known as Coronin3) is a member of the type 1 coronin that has been reported to regulate lamellipodial formation, cell migration, and membrane trafficking [12,13]. Coronin1C is ubiquitously expressed in various cells and localizes to the active sites of actin dynamics, including lamellipodia and membrane ruffles [14]. In some cancer cells, Coronin1C is involved in cell motility, invasion, and proliferation [15,16]. However, there are currently no studies on the role of Coronin1C in osteoclasts.

In this study, we found that the GDP form of Rab44 interacts with Coronin1C in RAW-D macrophages. Moreover, the knockdown of Coronin1C inhibits osteoclast differentiation through cell formation and fusion. Therefore, the decreased cell formation and fusion of Coronin1C-depleted osteoclasts might be due to the decreased migration of Coronin1C-knockdown macrophages.

## 2. Results

### 2.1. Rab44 Interacts with the Actin-Binding Protein, Coronin1C, in RAW-D Macrophages

To explore the binding protein(s) of Rab44, we performed a GTP-specific pull-down assay, which is a suitable method for identifying binding proteins to the GDP form of several GTPases, using murine RAW-D macrophages [17]. Figure 1a shows a schematic of the identification of a binding protein of Rab44. When Rab44 and its binding protein complexes were treated with ALP to stabilize the formation, the protein complexes were subsequently eluted with GTP. SDS-PAGE of the eluted proteins revealed that four additional bands were detected in GFP-Rab44 expressing (WT) cells compared with the GFP-expressing cells (Mock) (Figure 1b). Sequence analysis showed that the detected proteins were Prelamin-A/C (band #1), Coronin1C (band #2), IMP dehydrogenase (band #3), and Tropomyosin α3 chain (band #4) (Figure 1c). Of these proteins, we opted to focus on Coronin1C as its protein levels were the highest among the four proteins (see Figure 1b). Moreover, Coronin1C is detected by immunoprecipitation of mast cells as well as macrophage/osteoclasts. Therefore, we decided to study the role of Coroin1C in osteoclast differentiation.

### 2.2. Coronin1C Exclusively Binds to the GDP form of Rab44

To confirm whether GTP/GDP formation of the Rab domain is involved in the interaction of Rab44 and Coronin1C, we assessed the interaction between Rab44 and Coronin1C using RAW-D macrophages overexpressing GFP only as a control (mock), GFP-Rab44 (WT), a CA mutant of the Rab domain (Q596L), and a DN mutant of the Rab domain (T551N), as reported previously [18]. We initially tested the endogenous expression levels of Coronin1C in these cells. The highest protein level of endogenous Coronin1C was observed in DN mutant-expressing cells, although the expression level of Rab44 in the DN mutant-expressing cells was lower than those in WT and the CA mutant expressing cells (Figure 1d). Immunoprecipitation with an anti-GFP antibody revealed that the interaction of Coronin1C and GFP-Rab44 proteins was observed in WT and DN mutant-expressing cells, but not in the mock and CA-expressing cells (Figure 1d). The level of the DN mutant band was higher than that of the WT (Figure 1e), suggesting that Coronin1C exclusively interacts with the GDP form of Rab44.

Next, we confirmed the interaction between Rab44 and Coronin1C using PLA. Visible rolling circle amplification (RCA) products for the endogenous Rab44-Coronin1C complex were observed in the GFP-expressing cells (Mock) (Figure 2a). Under the same conditions, Rab44 WT-expressing cells displayed increased RCA products compared to mock cells (Figure 2a). Moreover, CA mutant-expressing cells showed significantly decreased RCA products compared with DN mutant-expressing cells (Figure 2a). A quantitative analysis of the number of RCA products per single cell is shown in Figure 2b. These results also indicate that Coronin1C binds exclusively to the GDP form of Rab44.

We also observed colocalization of native Coronin1C and Rab44 in bone marrow-derived macrophages (BMMs) three days after RANKL stimulation. Confocal microscopy revealed that Coronin1C partially colocalized with Rab44 (Figure 2c). These results indicate that Coronin1C colocalizes with Rab44 in RANKL-stimulated native BMMs and RAW-D macrophages.

### 2.3. Rab44 Constitutively Active (CA) Mutant Inhibits Osteoclast Differentiation, Whereas Dominant-Negative (DN) Mutant Enhances This Differentiation

Since Rab44 is upregulated by RANKL stimulation, we examined whether the binding partner, Coronin1C, was also increased by RANKL. We measured the expression levels of Coronin1C encoded by *CORO1C* in RAW-D macrophages after RANKL stimulation. QPCR analysis showed no significant difference in mRNA levels between RANKL-stimulated and unstimulated RAW-D cells, although the levels of *CORO1C* gradually decreased in both cells during cell culture (Figure 3a). We further examined the protein levels of Coronin1C after 3 days of RANKL stimulation. Western blot analysis of mock, WT, and CA mutants in unstimulated RAW-D cells indicated that the protein levels of Coronin1C in mock and WT-expressing cells were comparable. In contrast, the levels were slightly lower in CA mutant-expressing cells, and higher in DN mutant-expressing cells. (Figure 3b). Similar results were observed in RANKL-stimulated mock, WT, CA, and DN mutant-expressing cells (Figure 3b). Figure 3c shows qPCR results of Coronin1C in mock, WT, CA, and DN mutant-expressing cells on day 3 after RANKL stimulation. The mRNA expression data were consistent with the results of the western blot analysis. These results indicate that Coronin1C is not a RANKL-inducible protein, although Rab44 is upregulated by RANKL induction during osteoclast differentiation.

We assessed TRAP staining of WT, CA, and DN mutants expressing in RANKL-stimulated RAW-D macrophages (Figure 3d). The Rab44 WT and CA mutant expressing cells displayed almost all TRAP-negative cells (Figure 3d). In contrast, the Rab44 DN-expressing osteoclasts showed remarkably larger multinucleated formation than the mock cells (Figure 3d graph).

### 2.4. Knockdown of Coronin1C Inhibits Osteoclast Differentiation through Osteoclast Formation and Fusion

To investigate whether Coronin1C is implicated in osteoclast differentiation, we performed gene knockdown experiments using siRNA. Knockdown efficacy in RAW-D cells after 4 days of RANKL stimulation was confirmed by qPCR and Western blot analysis (Figure 4a,b). Compared to the control (mock) siRNA, #1, #2, and #3 diminished *CORO1C* mRNA expression levels showing the residual levels of approximately 50, 15 and 30%, respectively (Figure 4a). The coronin1C protein residuals were approximately 60, 60, and 30%, respectively (Figure 4b). TRAP staining revealed that the depletion of *CORO1C* by siRNA#2 reduced osteoclast differentiation compared to that in mock cells when cells were cultured for 4 days after stimulation with RANKL (Figure 4c). These results indicate that Coronin1C knockdown and CA mutant expression inhibit osteoclast differentiation, whereas DN mutant expression enhances osteoclast differentiation (Figure 3d and Figure 4c).

We also monitored the osteoclast formation between Coronin1C-knockdown and Mock in RANKL-induced RAW-D macrophages by phase-contrast microscopy. Coronin1C-knockdown osteoclasts impaired multinuclear formation, although the mock osteoclasts displayed a moderate multinuclear formation (arrowhead, Figure 4d). Movies of osteoclast formation of mock and Coronin1C-knockdown cells are provided (Appendix A). We further examined the mRNA levels of several osteoclast marker genes in mock and Coronin1C-knockdown osteoclasts (Figure 4e). Quantitative RT-PCR analysis revealed that mRNA levels of *CTSK* and *Src* were significantly lower in Coronin1C-knockdown osteoclasts compared to control cells (Figure 4e). Phalloidin staining, which stains F-actin, showed that Coronin1C-knockdown osteoclasts had less lamellipodia (Figure 4f arrowhead) and filopodia (Figure 4f light) formation than control cells (Figure 4f). Overall, these results indicate that Coronin1C regulates osteoclast differentiation through osteoclast formation and fusion in cells.

We further examined the effects of Coronin1C knockdown and Rab44 overexpression on osteoclast differentiation. We confirmed that Coronin1C knockdown efficiently suppressed the Coronin1C levels in GFP-Mock, WT, CA and DN-expressed cells (Figure 5a). Under these conditions, coronin1C knockdown reduced osteoclast formation in GFP-mock and DN-expressed cells (Figure 5b,c). These results indicate that Rab44 overexpression fail to compensate for Coronin 1C knockdown.

### 2.5. Coronin1C Knockdown Impairs the Migration and Chemotaxis of RAW-D Macrophages

As Coronin1C regulates cell migration in other cells, we monitored migration in RAW-D macrophages. Migration of Coronin1C-depleted RAW-D macrophages was decreased compared to that of mock cells (Figure 6a). The total cell migration distance of Coronin1C-knockdown RAW-D macrophages were significantly lower than those of mock cells (Figure 6b). Appendix A present the movies of migration of and Coronin1C-knockdown cells and the Mock cells. Similar results were observed concerning the chemotaxis of Mock and Coronin1C-knockdown RAW-D macrophages (Figure 7). Coronin1C depletion significantly diminished the number of migrated RAW-D macrophages even in response to the culture serum (without MCP-1) and MCP-1 (Figure 7). Taken together, the impaired migration of Coronin1C-depleted macrophages is probably implicated in the decreased cell formation and fusion of Coronin1C-depleted osteoclasts.

### 2.6. Coronin1C Knockdown Suppresses Osteoclastogenesis In Vivo

We finally examined the in vivo role of Coronin1C using a mouse model of calvarial osteoclast induction on the calvaria. The knockdown efficacy of native cells in the calvaria after 6 days of RANKL stimulation was confirmed by QPCR (Figure 8a). RANKL stimulation significantly induced osteoclast formation in the Mock mice compared to sham mice (Figure 7b,c). Under these conditions, Coronin1C knockdown significantly decreased osteoclastogenesis in mouse calvaria (Figure 8a–c). These results indicate that Coronin1C depletion also inhibits osteoclast formation in vivo.

## 3. Discussion

In this study, we demonstrated that Rab44 interacts with Coronin1C in RAW-D macrophages preferentially through its GDP form. Coronin1C partially colocalized with Rab44 in RANKL-stimulated native BMMs and RAW-D macrophages. Knockdown of Coronin1C inhibited osteoclast differentiation through osteoclast formation and fusion. Coronin1C knockdown impaired migration and chemotaxis in RAW-D macrophages. Thus, Coronin1C is likely to act as a GDP-specific Rab44 effector that controls osteoclast formation by regulating cell motility in macrophages.

Rab44 was found to interact with Coronin1C in RAW-D macrophages, which is the second example of Rab44 binding proteins. Our previous study indicated that Rab44 protein associates with vesicle-associated membrane protein 8 (VAMP8), a v-SNARE protein, in native mouse bone marrow mast cells as well as ectopic expression cells of rat basophilic leukemia-2H3 [18]. As VAMP8 interacts with wild-type Rab44 and its CA mutant in mast cells, VAMP8 might be a GTP-specific effector of Rab44 [18]. In contrast, the present study indicates that the GDP form of Rab44 predominantly interacts with Coronin1C in macrophages, suggesting that Rab44 negatively regulates the function of Coronin1C. This notion is consistent with the present findings that the CA mutant inhibited osteoclast differentiation, whereas the DN mutant enhanced such differentiation, and vice versa. (Figure 3d). Therefore, one of the mechanisms by which Rab44 negatively regulates osteoclast differentiation might be due to the interaction between the GDP-form Rab44 and Coronin1C.

The present finding that the GDP form of Rab44 interacts with Coronin1C in macrophages and osteoclasts is reminiscent of the previous results of the interaction between the GDP form of Rab27A and Coronin1C in pancreatic β cells [19,20]. During the transport of insulin granules in β cells, the GDP-form of Rab27A controls “endocytosis” by interacting with GDP-form effectors, such as Coronin1C [19] and IQGAP1 [21]; however, the GTP-form of Rab27A regulates “exocytosis” by interacting with GTP-form effectors, such as Exophilin8/Slac2-c/MyRIP [22], Granuphilin/Slp4a [23], and Exophilin7/JFC1/Slp1 [24]. Therefore, the switching mechanisms between the GTP and the GDP form of Rab44 may alter effector molecules, thereby controlling the movement of membrane trafficking.

In this study, Coronin1C depletion inhibited osteoclast differentiation through cell formation and fusion. According to previous studies, Coronin1A, a member of the type 1 coronin, has no effect on osteoclast differentiation, but it negatively regulates bone resorption and exocytotic secretion of the lysosomal protease cathepsin K [25]. As Coronin1A inhibits the lipidation of the autophagy-related protein, LC3, at the ruffled border, and impairs the fusion between lysosomes and the plasma membrane, Coronin1A might be involved in lysosomal fusion and the regulation of the secretion pathway in osteoclasts [25].

Several reports have shown that Cononin1C is implicated in protrusion formation, cell migration, and invasion in normal and cancer cells. For example, primary fibroblasts derived from Coronin1C-deficient mice display impaired cell migration and protrusion formation through abnormal actin filaments, microtubules, and intermediate filaments [26]. The involvement of Coronin1C in migration, invasion, and metastatic abilities has been observed in various cancer cells, such as human gastric cancer cells [15,27,28], melanoma cells [29], breast cancer cells [30,31], hepatocellular carcinoma cells [32], glioblastoma cells [33]. Thus, the actin-binding protein Coronin1C is commonly involved in cell motility, including migration, invasion, and chemotaxis in many cell types.

In conclusion, this study shows that Coronin1C is a GDP-specific Rab44 effector that controls osteoclast differentiation by regulating cell motility.

## 4. Materials and Methods

### 4.1. Antibodies and Reagents

Anti-GFP mAb-agarose (code. D153-8) and anti-GFP (Code. no. 598) were obtained from MBL (Nagoya, Japan). Mouse monoclonal anti-Coronin1C (Cat. no. sc-376919) was purchased from Santa Cruz Biotechnology (Santa Cruz, CA, USA). Anti-GAPDH (Cat. no. 2118S), Alexa Fluor 488 goat anti-rabbit IgG, and Alexa Fluor 555 goat anti-mouse IgG were purchased from Cell Signaling Technology (Danvers, MA, USA). Rab44 antibody was raised in rabbits using a recombinant protein and prepared as previously described [34]. Recombinant RANKL was prepared as described previously [35]. M-CSF was purchased from Kyowa Hakko Kogyo (Tokyo, Japan). 4′,6-diamidino-2-phenylindole (DAPI) and Alexa Fluor 488 Phalloidin were from Thermo Fisher (Waltham, MA, USA). Alkaline phosphatase (ALP) (Cat. no. 10713023001), and guanosine 5′-triphosphate sodium salt hydrate (GTP) (Cat. no. G8877), the protease inhibitor cocktail, Duolink in situ PLA probe anti-mouse MINUS, anti-rabbit PLUS, detection reagent red, and wash buffers A and B were from Sigma-Aldrich (Tokyo, Japan). MCP-1 was obtained from FUJIFILM Wako (Osaka, Japan). Trypsin Gold and Mass Spectrometry Grade (Cat. no. V5280) were purchased from Promega (Tokyo, Japan). Transwell 24-well plates were obtained from Corning, Inc. (Corning, NY, USA). SpongeCol^®^ (collagen sponge) was purchased from Advanced BioMatrix, (Carlsbad, CA, USA).

### 4.2. Cell Culture

Osteoclast differentiation was performed by culturing RAW-D cells, a subclone of the RAW264.7 mouse macrophage cell line, in the α-minimal essential medium (α-MEM) containing 10% fetal bovine serum with RANKL (100 ng/mL) at 37 °C in 5% CO_2_ for the indicated period. RAW-D cells were kindly provided by Prof. Toshio Kukita (Kyushu University, Japan) [36,37]. Bone marrow macrophages (BMMs) were isolated according to a previously described method [38]. Briefly, mouse femoral and tibial bone marrow cells were cultured overnight in α-MEM containing 10% fetal bovine serum (FBS) at 37 °C in 5% CO_2_ in the presence of M-CSF (50 ng/mL). Non-adherent cells were harvested and cultured in α-MEM containing 50 ng/mL M-CSF. After 72 h, adherent cells were collected as BMMs. BMMs were replated and further cultured for 72 h in the presence of M-CSF (30 ng/mL) and RANKL (100 ng/mL).

### 4.3. Retrovirus Construction and Expression of a Constitutively Active (CA) and a Dominant-Negative (DN) Mutants of Mouse Rab44 in RAW-D Cells

Mouse Rab44 wild-type expressing RAW-D cells were created as previously described [9]. The Rab44 CA and DN retrovirus vectors were produced by PCR from wild type one. The primers were used for

Rab44 CA forward: GCT GGA CTG GAG AGG TAC CAC AGC CTC

and reverse: CCT CTC CAG TCC AGC TGT GTC CCA CAG

Rab44 DN forward: GGC AAG AAC TCA TTC CTA CAC CTG CTA

and reverse: GAA TGA GTT CTT GCC CAC ATT GGA GTC

The cDNAs were amplified by PCR using PrimeStar MAX DNA polymerase (Takara Bio, Tokyo) with 35 cycles of denaturation at 98 °C for 10 s, annealing at 55 °C for 5 s, and extension at 72 °C for 45 s. The pMSCVpuro-GFP-vectors, kindly gifted by Prof. Kosei Ito (Nagasaki University, Japan), were used. To generate green fluorescent protein (GFP)-Rab44 fusion protein, the amplified fragments were fused with above vector using In-Fusion cloning kit (Clontech, Mountain View, CA, USA), then transfected into HEK293T cells by using the Lipofectamine3000 kit (Life Technologies, Gaithersburg, MD, USA), according to the manufacturer’s instructions. After incubation at 37 °C in 5% CO_2_ for 48 h, the retrovirus-containing supernatants were collected and used to infect RAW-D cells. Rab44 overexpressing cells were selected by puromycin (3 μg/mL) in α-MEM. Every 3 days, change medium change was performed. Approximately two weeks later, several cloned cells were obtained.

### 4.4. GTP-Specific Pull-Down Assay

Cells were rinsed twice with ice-cold PBS and lysed in a cell lysis buffer (50 mM Tris–HCl pH 8.0, 1% Nonidet P-40, 0.5% sodium deoxycholate, 0.1% SDS, 150 mM NaCl, 1 mM PMSF, and proteinase inhibitor cocktail). ALP (25 units) was added to the lysate and incubated on ice for 1 h. Thereafter, anti-GFP monoclonal antibody (mAb) agarose was added, and the mixture was placed in a rotator and reacted overnight at 4 °C. After five washes with PBS, GTP (5 mM) was added to the agarose complex and incubated on ice for 5 min. Thereafter, the supernatant was collected by centrifugation (12,000 rpm, 5 min), mixed with 4× SDS sample buffer (0.25 M Tris-HCl pH 6.8, 8% SDS, 28% glycerol, 0.4 M DTT, 1% protease inhibitor cocktail), and boiled at 100 °C for 5 min.

### 4.5. Western Blotting Analysis

Western blotting was performed as described previously [39]. Cells were rinsed twice with ice-cold PBS and lysed in a cell lysis buffer, as described in the GTP-specific pull-down assay. Equal amounts of protein (5 μg) were applied to each lane. After sodium dodecyl sulfate-polyacrylamide gel electrophoresis (SDS-PAGE), the proteins were electroblotted onto a polyvinylidene difluoride membrane. The membranes were blocked with 5% nonfat milk solution containing Tris-buffered saline (TBS)/0.1% Tween 20 for 1 h at room temperature, probed with various antibodies overnight at 4 °C, washed, incubated with horseradish peroxidase-conjugated secondary antibodies (Cell Signaling Technology, Danvers, MA, USA), and finally detected with Immobilon Forte (Millipore, Burlington, MA, USA). Immunoreactive bands were analyzed using LAS4000-mini (FUJIFILM Wako, Tokyo, Japan).

### 4.6. Matrix Associated Laser Desorption/Ionization (MALDI)-Time of Flight Mass Spectrometry (TOF-MS)

Samples from the pull-down assay were subjected to SDS-PAGE and Coomassie Brilliant Blue (CBB) staining. Then, the band was removed, reacted with the decolorizing solution (50% CH_3_CN, 25 mM NH_4_HCO_3_), dehydrated with 100% CH_3_CN, and mixed with the reducing solution (10 mM DTT, 25 mM NH_4_HCO_3_). The products were subjected to an alkylation treatment (55 mM iodoacetamide, 25 mM NH_4_HCO_3_), dehydrated, and then treated with trypsin (25 ng/μL) to extract the peptide. The extracted peptides were mixed with a matrix (1 mg α-cyano-4-hydroxy-cnnamic acid in 100 μL of 50% CH_3_CN, 0.1% TFA), applied to a target plate, and analyzed using a MALDI-TOF device (Ultraflex III, Bruker, MA, USA). The calibration peak was obtained using a peptide calibration standard (Bruker, MA, USA). The analyzed data were subjected to Mascot Search, which is a bioanalysis tool to search for proteins.

### 4.7. Proximity Ligation Assay (PLA)

The proximity ligation assay was performed according to the manufacturer’s protocol using Duolink in Situ PLA probes and red detection reagents. Briefly, cells were fixed with 4% paraformaldehyde (PFA), permeabilized, and blocked as described for immunocytochemistry. The cells were then incubated overnight at 4 °C with the two primary antibodies, rabbit anti-Rab44 and mouse anti-Coronin1C diluted with Duolink Antibody Diluent. After washing with wash buffer A, the cells were incubated with anti-mouse MINUS and anti-rabbit PLUS probes at 37 °C for 1 h. Thereafter, the cells were washed with buffer A and reacted with ligase in ligation buffer at 37 °C for 30 min. After washing with buffer A, the cells were incubated with polymerase in amplification buffer at 37 °C for 100 min in a preheated humidity chamber. The cells were then washed with buffer B and 0.01 X buffer B. The cells were incubated with DAPI for 30 min to stain the nuclei, and then washed with PBS. PL signals were observed using a confocal microscope (LSM800; Carl Zeiss, AG, Jena, Germany). To quantify the results, four images containing approximately 100 cells were taken with a 40× objective lens using a confocal microscope and three independent experiments were performed. The PL particles and nuclei of each image were counted using the ImageJ software. The total number of PL particles in the image was divided by the number of nuclei.

### 4.8. Quantitative Real-Time Polymerase Chain Reaction (QPCR) Analysis

Total RNA was extracted using TRIzol reagent (MRC, Cincinnati, OH, USA). Reverse transcription was performed using oligo(dT) 15 primer (Promega, Madison, WI, USA) and Revertra Ace (Toyobo, Osaka, Japan). Quantitative RT-PCR was performed using a LightCycler 480 (Roche Diagnostics, Mannheim, Germany). cDNA was amplified using Brilliant III Ultra-Fast SYBR Green QPCR Master Mix (Agilent Technology, La Jolla, CA, USA). The following primer sets were used:

*GAPDH* forward, AAATGGTGAAGGTCGGTGTG reverse, TGAAGGGGTCGTTGATGG.

*CORO1C* forward, AACGGTAGTCTCATCTGCAC reverse, AGTAGATGATGCTGGTGTCC.

*DCSTAMP*, forward: CTAGCTGGCTGGACTTCATCC and reverse: TCATGCTGTCTAGGAGACCTC;

*OCSTAMP* forward: TGGGCCTCCATATGACCTCGAGTAG and reverse: TCAAAGGCTTGTAAATTGGAGGAGT;

*CTSK* forward: CAGCTTCCCCAAGATGTGAT and reverse: AGCACCAACGAGAG-GAGAAA;

*Src* forward: AGAGTGCTGAGCGACCTGTGT and reverse: GCAGA-GATGCTGCCTTGGTT.

### 4.9. Immunocytochemistry

Immunocytochemistry was performed as described previously [34,40]. The cells were cultured on cover glasses and fixed with 4% PFA for 20 min at room temperature. The fixed cells were then washed three times with PBS for 10 min and permeabilized with 0.2% Triton X-100 in PBS for 15 min. The cells were blocked with 0.2% gelatin for 1 h and subsequently incubated overnight at 4 °C with primary antibodies. Thereafter, the cells were washed three times with PBS for 10 min and then stained by secondary antibodies, such as Alexa Fluor 488 goat anti-rabbit IgG or Alexa Fluor 555 goat anti-mouse IgG. Nuclear staining with DAPI and F-actin staining with Phalloidin were also performed. The samples were subjected to microscopy using a laser-scanning confocal imaging system (LSM800; Carl Zeiss AG, Jena, Germany).

### 4.10. Small Interfering RNA (siRNA)

siRNA experiments with RAW-D cells were performed as previously described [9]. The target sequences of murine CORO1C siRNA were as follows: #1 sense GAGAAAGUGCGAGCCCAUU anti-sense AAUGGGCUCGCACUUUCUC #2 sense CAGUCAAGACGAGCGCAUU, anti-sense AAUGCGCUCGUCUUGACUG #3 sense GACAUUCCAUGUCAGAUCU anti-sense AGAUCUGACAUGGAAUGUC. RAW-D cells seeded in dishes or plates were cultured in antibiotic-free media for 1 d. The next day, the siRNA was transfected into RAW-D cells using Lipofectamine RNAiMAX™ transfection reagent (Invitrogen, Carlsbad, CA, USA) according to the manufacturer’s instructions. The cells were incubated with 10 pmol of siRNA for 24 h and then used for each experiment.

### 4.11. Tartrate-Resistant Acid Phosphatase (TRAP) Staining

TRAP staining was performed as previously described [41]. Briefly, cells were fixed with 4% PFA at room temperature for 40 min and then treated with 0.2% Triton X-100 in PBS at room temperature for 10 min. Finally, the cells were incubated with 0.01% naphthol AS-MX phosphate (Sigma-Aldrich, Tokyo, Japan) and 0.05% fast red violet LB salt (Sigma-Aldrich, Tokyo, Japan) in the presence of 50 mM sodium tartrate and 90 mM sodium acetate (pH 5.0) for TRAP activity. TRAP-positive cells with three or more nuclei were considered to be mature osteoclasts.

### 4.12. Chemotaxis Assay

A chemotaxis assay was performed as described previously [42]. Briefly, the migration of cells (2 × 10^5^ cells) in the presence or absence of MCP-1 was assessed in a Transwell 24-well plate with a polyethylene terephthalate membrane (pore size 8 μm). Cells were loaded into the upper chambers, and the lower chambers were filled with or without MCP-1(0.1 nM). The plate was incubated at 37 °C in 5% CO_2_ for 90 min or 240 min. The cells that migrated to the underside of the membrane were fixed with 4% paraformaldehyde for 10 min, subsequently stained with hematoxylin for 30 min, and then washed with PBS. The cells on the upper side membrane were removed using a swab, and the migrated cells were counted using an optical microscope.

### 4.13. Live Cell Imaging

Live cell imaging was performed by the method as previously described [35]. Briefly, after cells were seeded in a 6-well plate, the plate was transferred to a time-lapse microscope containing a 37 °C, 5% CO_2_ incubation chamber for 2 h to allow cells to settle and for the plastic ware to equilibrate. Images were obtained using an inverted Real-Time Cultured Cell Monitoring System CCM-1.4Z (ASTEC, Fukuoka, Japan) with a 10 × objective lens and bright field channels at 10-min intervals up to 12 h for macrophage cells and 10-min intervals up to 72 h for osteoclasts. Cell tracking analysis for migration was performed using the manual tracking plugin of ImageJ software.

### 4.14. In Vivo Experiments with Mouse Calvaria

Seven-week-old male C57BL mice were obtained from CLEA Japan (Tokyo, Japan). Mouse calvaria were implanted with collagen sponges pretreated with mock or *CORO1C* siRNA (12 μL of 50 μM solution) mixed with Lipofectamine RNAiMAX (10 μL; Invitrogen). Sham mice underwent only periosteal avulsion of the calvarial bone. The next day, RANKL (2 mg/kg) solution was added to the collagen sponges. Solutions of mock or CORO1C siRNA and RANKL were injected at three-day intervals. The calvariae were sacrificed on day 6 for mRNA collection and TRAP staining.

### 4.15. Animal Experiments and Care

All experiments and care were performed in our facilities using protocols approved by Nagasaki University Animal Care Committee. (Approval Number: 2107211733-3).

## Figures and Tables

**Figure 1 ijms-23-06619-f001:**
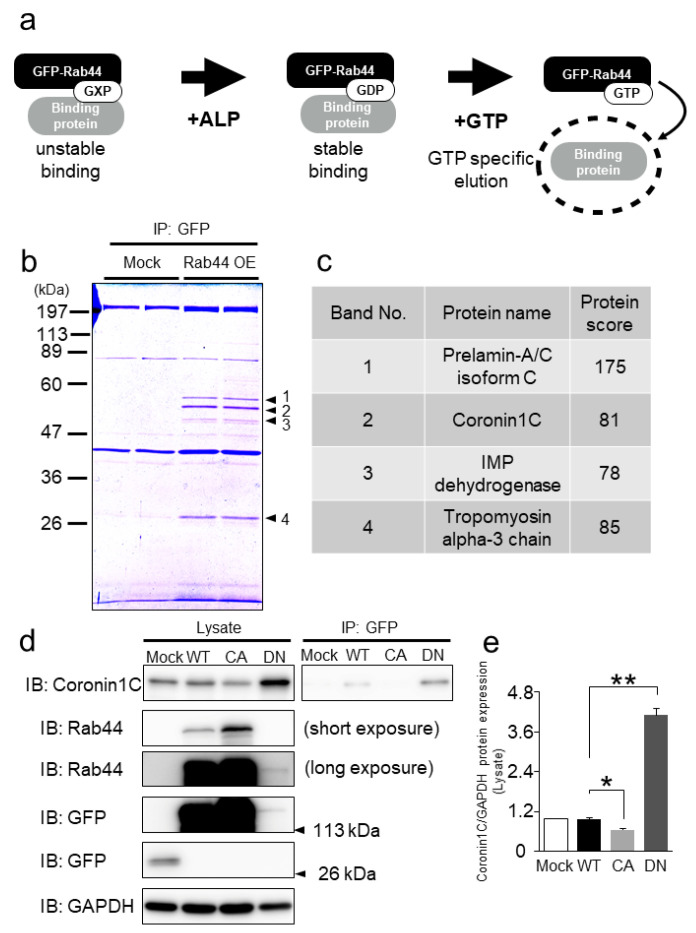
Identification of Coronin1C as a GDP-specific Rab44-interacting protein in mouse macrophage RAW-D cells. (**a**) Schematic of GDP-bound Rab44-interacting proteins. Rab44 binds to and dissociates from GEF in the presence of GTP or GDP (GXP) in cellular lysates (unstable binding). The addition of ALP (25U) removed GXP and maintained the binding between GEF and Rab44 (stable binding). When GTP (5 mM) was added, GDP-specific Rab44 interacting proteins were extracted (GTP-specific elution). (**b**) CBB staining of the GDP-bound Rab44-interacting proteins. Immunoprecipitation (IP) was performed using beads with GFP antibody in RAW-D cells expressing GFP protein only (mock) or Rab44 tagged with GFP (Rab44 OE). The eluates (same protein amounts) were subjected to SDS-PAGE detected by CBB staining. (**c**) The four proteins were identified by MALDI-TOF-MS using Mascot Search. Prelamin-A/C was detected as band #1, Coronin1C was band #2, IMP dehydrogenase was band #3, and Tropomyosin α3 chain was band #4. Protein scores greater than 65 were considered significant (*p* < 0.05). (**d**) Western blot analysis of immunoprecipitation (IP) experiments with GFP antibody in RAW-D cells expressing GFP protein only (Mock), Rab44 tagged with GFP (WT), a constitutively active (CA) mutant (Q596L), and a dominantly negative (DN) mutant (T551N). Cell lysates or IP samples using a GFP antibody were subjected to SDS-PAGE followed by Western blotting with antibodies against Coronin1c, Rab44, GFP, and GAPDH. GAPDH was used as the loading control. (**e**) Quantitative analysis of proteins in cell lysates by relative chemiluminescence intensity measured with ImageJ. Measurements were normalized to those of mock without RANKL stimulation. * *p* < 0.05, ** *p* < 0.01; compared to WT with or without RANKL stimulation, respectively.

**Figure 2 ijms-23-06619-f002:**
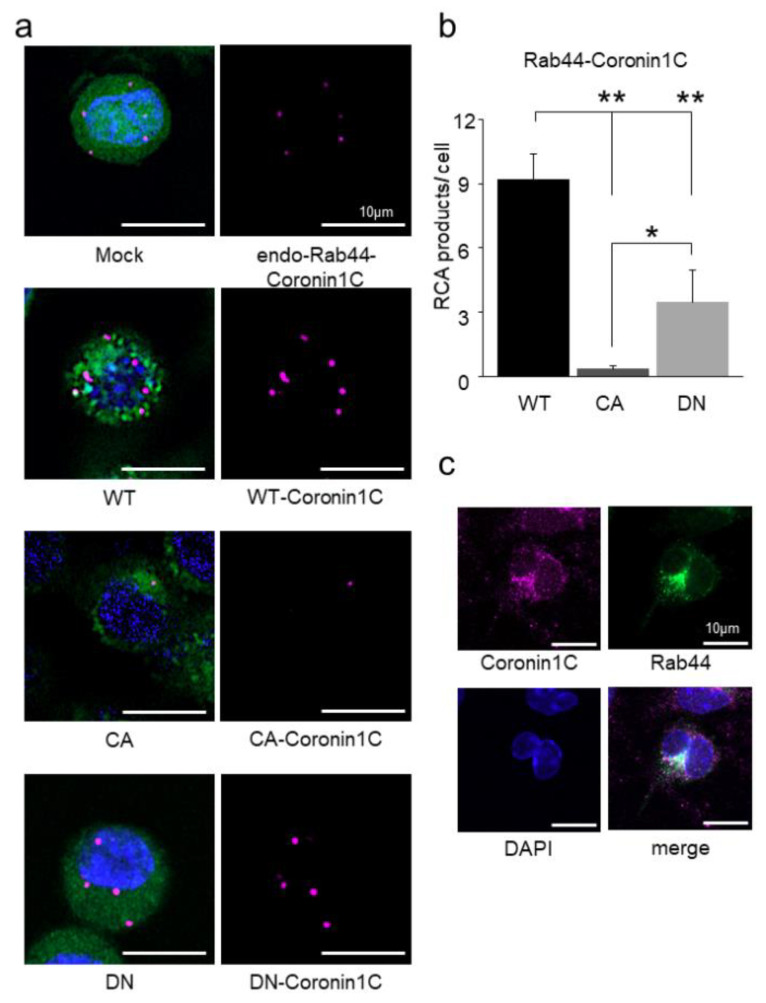
Proximity ligation assay of the interaction between Rab44 and Coronin1C. (**a**) Detection of proximity ligation signals between Rab44 and Coronin1C in RAW-D cells using a confocal microscope. GFP proteins in Rab44 (WT, CA, DN) with GFP tag expressing cells were fluorescently detected as green, rolling circle amplification (RCA) products of Rab44 and Coronin1C were detected as magenta, and DAPI was detected as Blue (nuclei). Bar: 50 μm. (**b**) Quantitative analysis of the number of the RCA products in a cell using ImageJ software. Data are expressed as mean ± SD (*n* = 4). * *p* <0.05, ** *p* <0.01. (**c**) Localization of Rab44 and Coronin1C in bone marrow-derived macrophages (BMMs) using confocal microscopy. BMMs were cultured in M-CSF (30 ng/mL) and RANKL (100 ng/mL) for three days, followed by immunofluorescent staining with Rab44 and Coronin1c antibodies. Coronin1C (Magenta), Rab44 (Green), and the nuclei (Blue).

**Figure 3 ijms-23-06619-f003:**
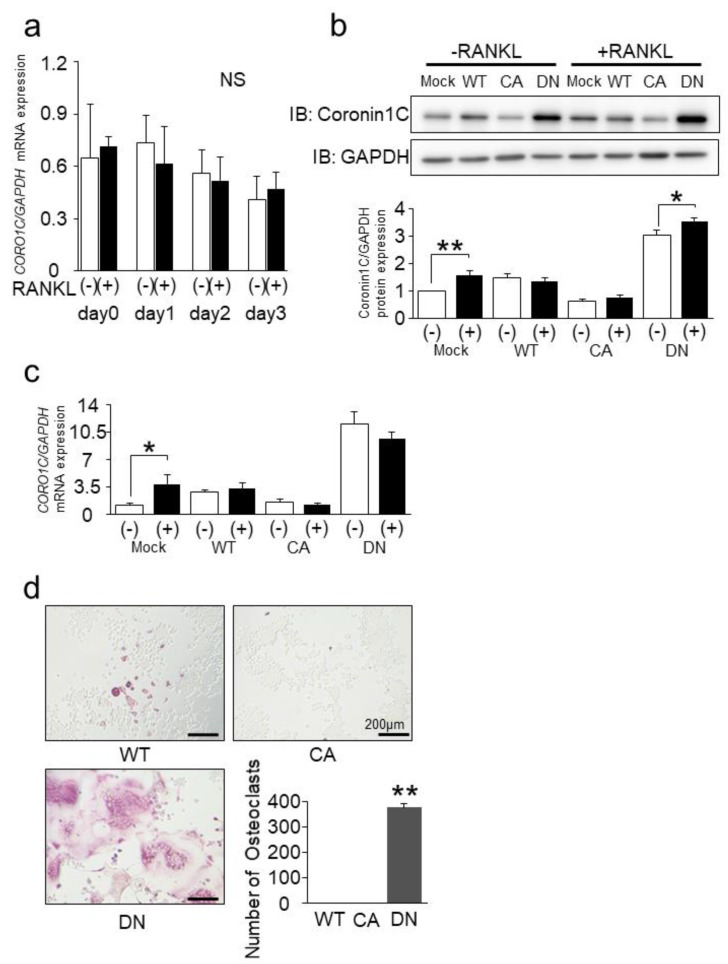
Expression and localization of Coronin1c during osteoclast differentiation. (**a**) qPCR analysis of *CORO1C* mRNA expression levels in RANKL stimulated RAW-D cells during osteoclast differentiation. The data are expressed as the mean ± SD of the values from three independent experiments. No significant differences were found. (**b**) RAW-D cells expressing mock, WT, and CA were cultured in the presence or absence of RANKL (100 ng/mL) for 3 days. Cell lysates from the expressing RAW-D cells were subjected to SDS-PAGE followed by Western blotting with antibodies against Coronin1C, and GAPDH. GAPDH was used as the loading control. Quantitative analysis of proteins in cell lysates by relative chemiluminescence intensity measured with ImageJ. Measurements were normalized to those of mock without RANKL stimulation. * *p* < 0.05, ** *p* < 0.01; compared to Mock with or without RANKL stimulation, respectively. (**c**) The mRNA expression levels of Coronin1C in mock, WT, CA expressed RAW-D cells after treatment with or without RANKL for 3 days were analyzed by qPCR. * *p* < 0.05; compared with the mock cells. (**d**) TRAP staining of multinucleated cells derived from WT, CA-mutant, and DN-mutant expressing cells treated with RANKL for 3 days. Bar: 200 μm. The number of TRAP-positive multinucleated osteoclasts was counted. ** *p* < 0.01; compared to WT-expressing cells.

**Figure 4 ijms-23-06619-f004:**
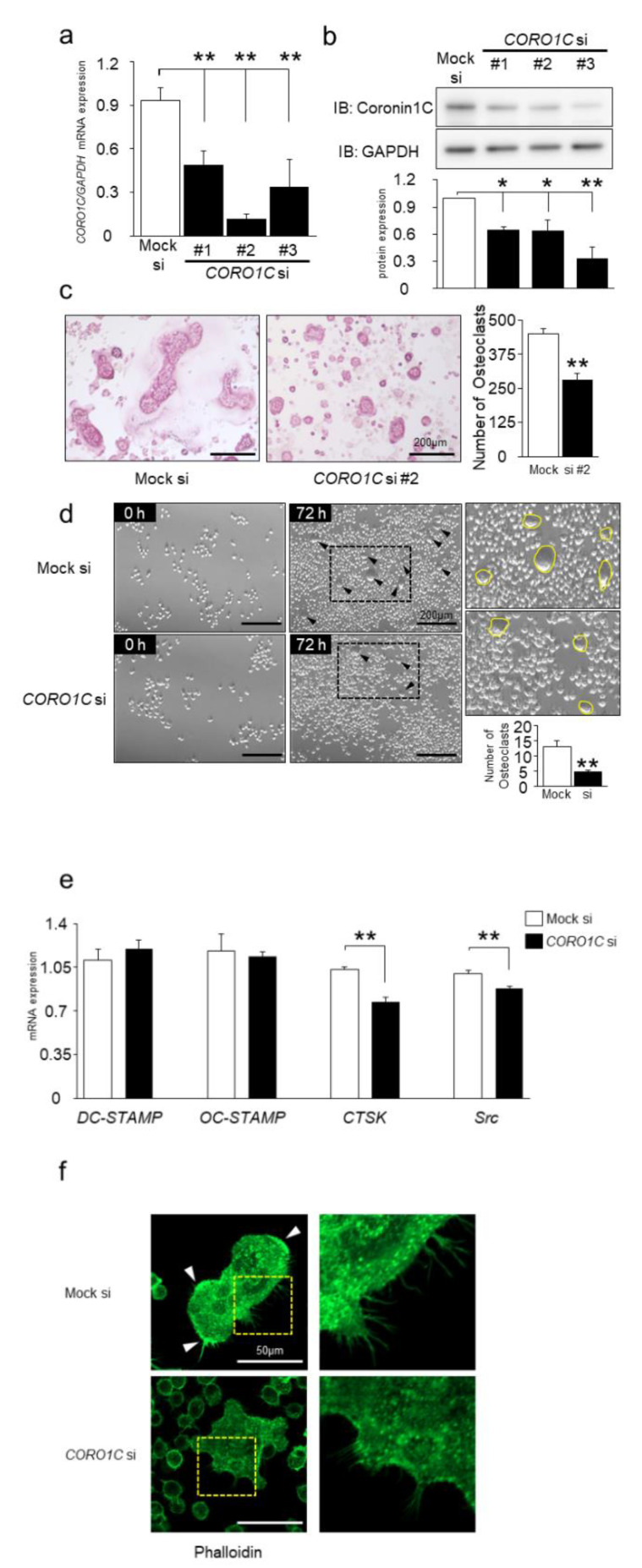
Effects of Coronin1C knockdown on osteoclast differentiation. (**a**) For knockdown efficiency, RAW-D cells were transfected with 3 types of *CORO1C* siRNA (Coronin1C si), followed by stimulation with RANKL (100 ng/mL) for 3 days. The Coronin1C mRNA levels were analyzed by qPCR. ** *p* < 0.01; compared with the mock cells. (**b**) The Coronin1C protein levels of 3 types were analyzed by Western blot. Quantitative analysis of proteins in cell lysates by relative chemiluminescence intensity measured with ImageJ. Measurements were normalized by Mock si. * *p* < 0.05, ** *p* < 0.01; compared to Mock si. (**c**) TRAP staining of multinucleated cells derived from Coronin1C si and the Mock si cells treated with RANKL for 4 days. ** *p* < 0.01; (**d**) Time-lapse images at 0 h and 72 h of Mock si (upper) and *CORO1C* si cells (lower). The arrowheads point to multinucleated cells. The magnified images on the right are images of the area circled by the dotted line in the 72-h images. Cells bordered in yellow are multinucleated cells. The number of multinucleated cells was counted. ** *p* < 0.01; compared to mock si cells. (**e**) Comparison of mRNA levels of various osteoclast marker genes in *CORO1C* si and the Mock si cells. RAW-D cells were cultured with RANKL (100 ng/mL) for 3 days. After mRNA isolation from these cells, RT-PCR was performed. ** *p* < 0.01, compared with the Mock si cells. (**f**) Phalloidin staining of multinucleated cells derived from Coronin1C si and the Mock si cells treated with RANKL for 3 days. The arrowheads point to lamellipodia in cells. The area enclosed by the yellow dotted line has been enlarged and is shown to the right. The enlarged photos show filopodia.

**Figure 5 ijms-23-06619-f005:**
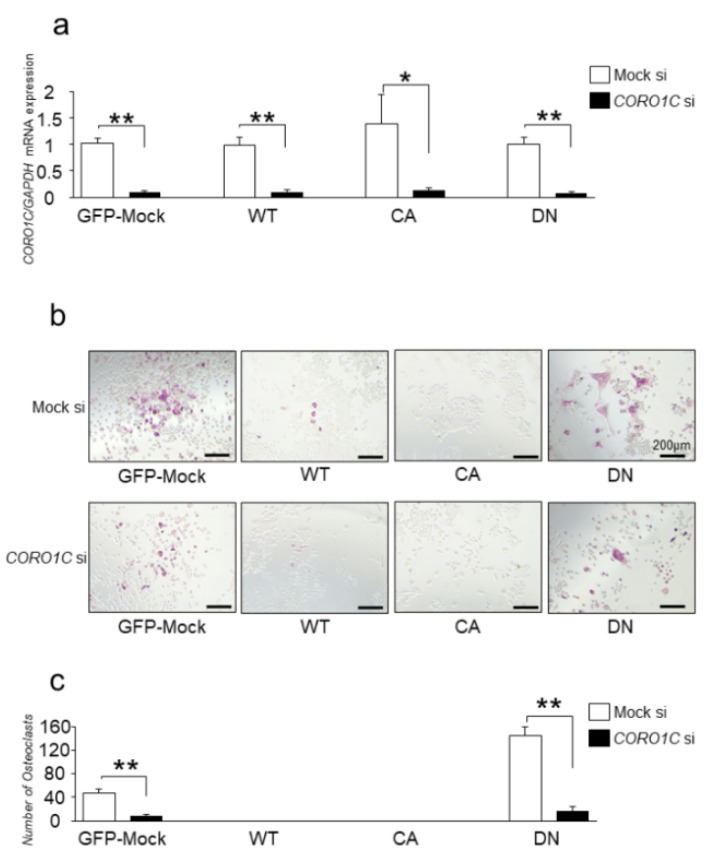
Effects of Coronin1C knockdown and Rab44 overexpression on osteoclast differentiation. (**a**) GFP-Mock, WT, CA and DN expressed RAW-D cells were transfected with *CORO1C* siRNA (Coronin1C si), followed by stimulation with RANKL (100 ng/mL) for 3 days. The Coronin1C mRNA levels were analyzed by qPCR. * *p* < 0.05; ** *p* < 0.01; compared with the mock cells. (**b**) TRAP staining of the GFP-mock, WT, CA and DN expressed RAW-D cells transfected with Coronin1C si and the Mock si cells after stimulation with RANKL for 3 days. (**c**) The number of TRAP-positive multinucleated osteoclasts, which contained three or more nuclei, was counted. ** *p* < 0.01; compared to mock cells.

**Figure 6 ijms-23-06619-f006:**
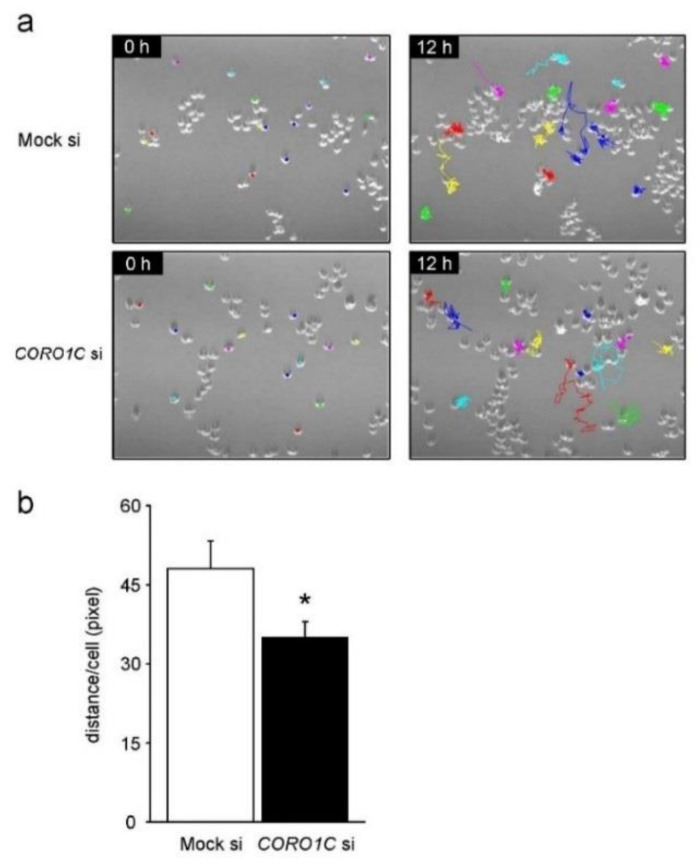
Migration of the mock and Coronin1C-knockdown RAW-D cells. (**a**) Mock and Coronin1C-knockdown RAW-D cells seeded onto a 6-well plate were analyzed by time-lapse video microscopy. Micrographs of time-lapse imaging showing cell tracks. Representative plots of approximately 15 cells of mock and Coronin1C-knockdown RAW-D cell migration tracks for a total duration of 12 h/track. The data and pictures are representative of three independent experiments. (**b**) The distance traveled between positions (path length) of 100 cells. * *p* < 0.05, compared with the mock cells.

**Figure 7 ijms-23-06619-f007:**
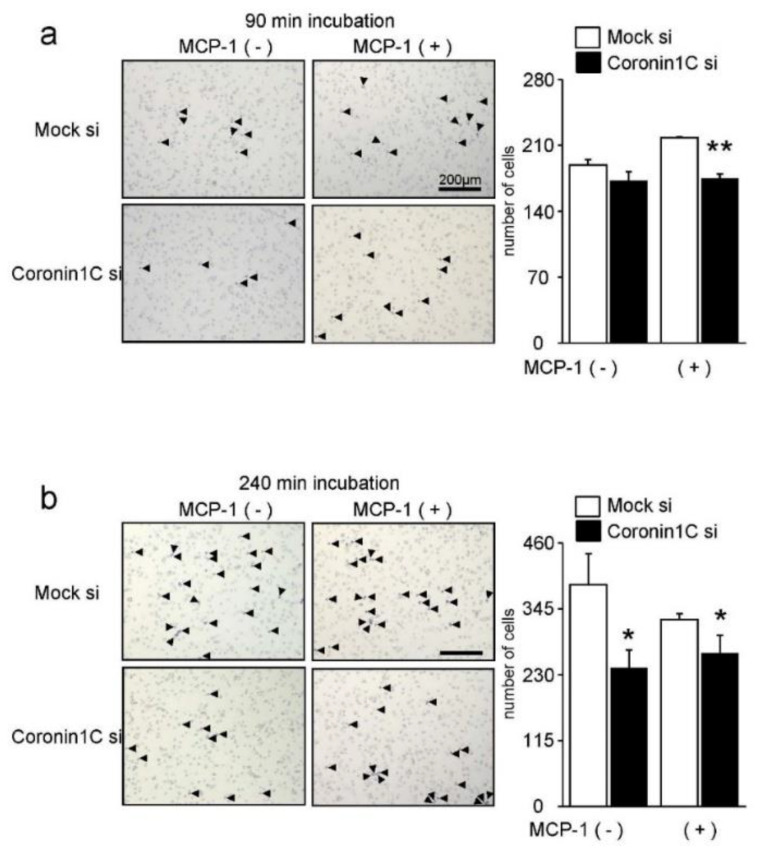
Chemotaxis of the Mock and Coronin1C-knockdown RAW-D cells. Chemotaxis was evaluated using a Transwell chamber. The cells in the 24-well chambers were treated with or without MCP-1 (0.1 nM) at 37 °C for 90 min (**a**) or 240 min (**b**). The cells that migrated from the upper to the lower well were fixed and stained with hematoxylin solution. The number of cells was counted by light microscopy. The arrowheads indicate cells. * *p* <0.05, ** *p* < 0.01; compared with the mock si.

**Figure 8 ijms-23-06619-f008:**
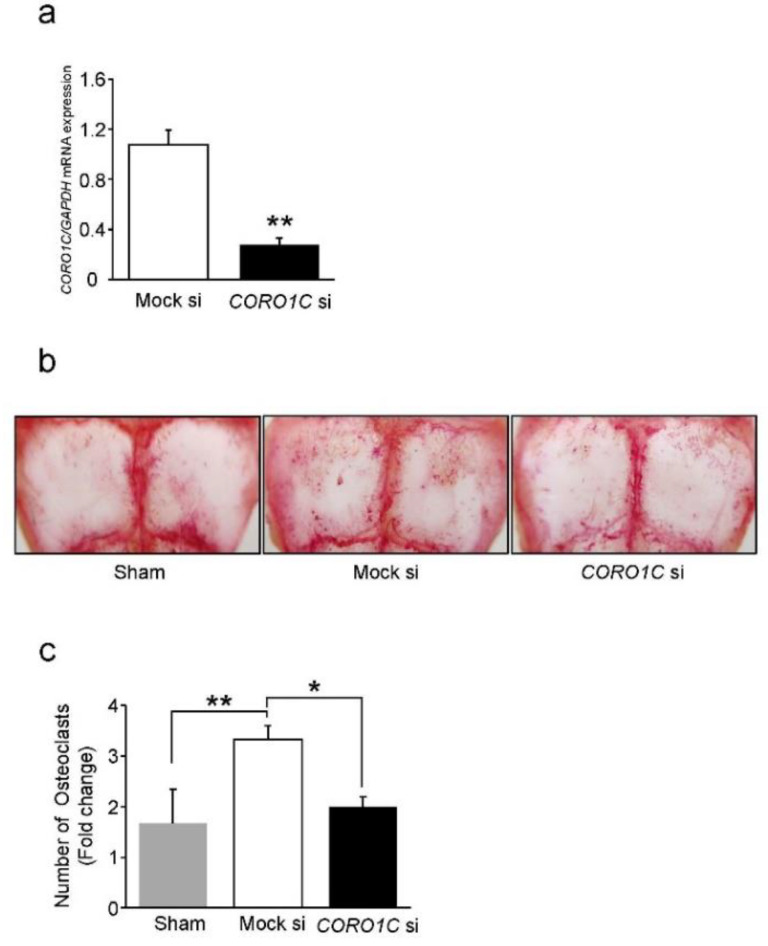
Effects of Coronin1C knockdown on osteoclastogenesis in vivo. Mouse calvaria were implanted with collagen sponges that were pretreated with mock or *CORO1C* siRNA. Sham mice underwent only periosteal avulsion of the calvaria bone. The next day, RANKL was added to the collagen sponges. The calvaria were sacrificed on day 6 for mRNA collection (**a**) and TRAP staining (**b**,**c**). (**a**) The *CORO1C* mRNA levels were analyzed by qPCR. ** *p* < 0.01; compared with the mock si. (**b**) TRAP-stained images of multinucleated cells induced to Sham and Mock si, *CORO1C* si calvaria treated with RANKL for 6 days. (**c**) The number of TRAP-positive cells was counted and normalized to Sham. * *p* < 0.05, ** *p* < 0.01 (*n* = 3).

## Data Availability

The data supporting the findings of this study are available in the figures and Appendix A of this article.

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
