# Peer review of "Coronin1C Is a GDP-Specific Rab44 Effector That Controls Osteoclast Formation by Regulating Cell Motility in Macrophages"

_ijms, 2022, doi:10.3390/ijms23126619_

Round 1

Reviewer 1 Report

The authors investigate the molecular mechanisms by which Rab44 negatively regulates osteoclast differentiation. They found that the GDP form of Rab44 interacted with the actin-binding protein, Coronin1C, in murine macrophages. The interaction of Rab44 with Coronin1C occurred in wild-type (WT) and a dominant-negative (DN) mutant of Rab44, but not in a constitutively active (CA) mutant form of Rab44. Consistent with these findings, the expression of the CA mutant inhibited osteoclast differentiation, whereas that of the DN mutant enhanced this differentiation. Coronin1C-knockdown osteoclasts impaired multinuclear formation, migration, and chemotaxis.

Major points:

1) As reported by the authors in a previous article, Rab44 is upregulated after 2-3 days of stimulation with RANKL. Rab44 overexpression prevents osteoclast differentiation, and Rab44 knockdown instead promotes RANKL-induced differentiation of osteoclasts. Mechanically, Rab44 influences NFATc1 expression and RANKL-RANK signaling (Ca+2, DC-STAMP, CtsK, MMP9) (Yu Yamaguchi et al. 2018). Here the authors reported that both Rab44  (WT) and  DN-mutant overexpression bind Coronin1c and the latter increases osteoclast differentiation (TRAP assay).

1) In light of the above It should be tested the effects of Rab44 overexpression (WT) on osteoclast differentiation (RANKL-RANK signaling, NFATc1, DC-STAMP, CtsK, MMP9 expression).

2) Lanes 77-79 (fig1b.) Four proteins are detected in GFP-Rab44 expressing (WT) cells compared with the GFP-expressing cells (Mock).

2) Is it the expression of Coronin1c protein dependent on the Rab44 overexpression?

3) Lanes 88-92. The DN mutant of the Rab domain (T551N) used in this study has not been characterized before (see Tomoko Kadowaki et al. 2021). Is it the short or long form? As these forms are different in that the short one lacks the dominance that binds Calcium.

4) Lanes 93-96 (Fig. 1d). The highest protein level of endogenous Coronin1C was observed in DN mutant-expressing cells, although the expression level of Rab44 in the DN mutant-expressing cells was lower than those in WT and the CA mutant expressing cells

4) How could the authors explain this large difference in Rab44 expression level between mutant DN and WT or CA?

5) Lanes 96-98 (Fig 1d). Immunoprecipitation with an anti-GFP antibody revealed that the interaction of Coronin1C and GFP-Rab44 proteins was observed in WT and DN mutant-expressing cells, but not in the mock and CA-expressing cells.

5) How could the authors explain that in mock there is no interaction between Rab44 and Coronin1c?

6) (lanes 98-100). The level of the DN mutant band was higher than that of the WT, suggesting that Coronin1C exclusively interacts with the GDP form of Rab44

6) The Coronic1c protein expression level in the DN mutant was also higher than that of the WT in the lysate (as in IP-GFP). The authors should show a quantization of the Coronin1c protein level also in the lysates

7) Lanes 121-126 (figs 2a and 2b). Quantitative analysis shows that Coronin1c binds with Rab44 WT and DN-mutant, and the highest value is in WT cells, differently from what reported in fig. 1d. How do the authors explain this?

8) The authors measured Coronin1C expression levels in RAWD cells with or without RANKL addition during 0-3 days (Figure 3a) and found no significant differences. Then, they studied the expression level of the Coronin1c protein (fig 3b) and mRNA (fig 3c) after 3 days of stimulation with RANKL in Mock, WT, DN and CA

8) However, looking at the Mock sample with or without RANKL in Figure 3b, it appears that the expression level of the Coronin1c protein increases. To find differences in osteoclastogenesis, authors should compare the expression levels of the Coronin1c protein and mRNA (WB densitometry, qPCR) by analyzing the same sample / treatment in the presence / absence of RANKL (i.e., Mock-RANKL with Mock + RANKL , WT-RANKL with WT + RANKL, etc.). This way there are differences in the Mock and mutant DN samples.

9) How do the authors explain the big difference of Coronin1c protein expression level in Mock and DN samples or in the WT and DN samples (Figure 3b)?

10) In figure 3d the upper and lower panels don't look as the same magnification (Mock-CA and Mock-DN).

11) In figure 3d the authors analyzed osteoclast formation by comparing CA with Mock and DN with Mock. However, in Figure 1d they demonstrated binding of Coronin1c to Rab44 in WT and DN samples, but not in Mock and CA (-RANKL). Hence, the authors should analyze osteoclast formation in CA and DN-mutant versus overexpressed Rab 44 (WT).

12) Lanes 163-165 (Fig. 3d and 3e). In contrast, the Rab44 DN-expressing osteoclasts showed remarkably larger multinucleated formation than the Mock cells

12) Why in this study, the Rab44 DN-expressing osteoclasts are larger multinucleated whereas in the study of 2018 overexpression of GFP-Rab44 impaired osteoclastogenesis? In the figure 2b the the Rab44 WT-expressing sample is the best binding Coronin1c, why the authors did not use this as control instead Mock?

13) Compared to the control (mock) siRNA, #1, #2, and #3 diminished CORO1C mRNA expression levels by approximately 50%, 15%, and 30%, respectively (Fig. 4a). Coronin1C protein levels were reduced by approximately 60, 60, and 30%, respectively (Fig. 4b). (lanes 193-196)

13) Lanes 193-196. Compared to the control (mock) siRNA, #1, #2, and #3 diminished CORO1C mRNA expression levels by approximately 50%, 15%, and 30%, respectively  (Fig. 4a). Coronin1C protein levels were reduced by approximately 60, 60, and 30%, respectively (Fig. 4b).

13) I believe that if you want to indicate in the text that the expression levels diminished by approximately 50, 15 and 30% respectively, in the figure (fig 4a) you must indicate the residuals of the expression level (50, 85 and 70%), or you must write in the text that the residual levels of expression are 50, 15 and 30%. The same for protein expression (Figure 4b).

14) Why don't the authors knockdown Coronin1c in RAWD cells GFP-Mock, WT, CA and DN Rab44 expressing cells to see the combined effects of the overexpression of Rab44 active and inactive form with the binding to the Coronin1c protein?

Author Response

IJMS comments

Comments and Suggestions for Authors

The authors investigate the molecular mechanisms by which Rab44 negatively regulates osteoclast differentiation. They found that the GDP form of Rab44 interacted with the actin-binding protein, Coronin1C, in murine macrophages. The interaction of Rab44 with Coronin1C occurred in wild-type (WT) and a dominant-negative (DN) mutant of Rab44, but not in a constitutively active (CA) mutant form of Rab44. Consistent with these findings, the expression of the CA mutant inhibited osteoclast differentiation, whereas that of the DN mutant enhanced this differentiation. Coronin1C-knockdown osteoclasts impaired multinuclear formation, migration, and chemotaxis.

Major points:

1) As reported by the authors in a previous article, Rab44 is upregulated after 2-3 days of stimulation with RANKL. Rab44 overexpression prevents osteoclast differentiation, and Rab44 knockdown instead promotes RANKL-induced differentiation of osteoclasts. Mechanically, Rab44 influences NFATc1 expression and RANKL-RANK signaling (Ca+2, DC-STAMP, CtsK, MMP9) (Yu Yamaguchi et al. 2018). Here the authors reported that both Rab44 (WT) and DN-mutant overexpression bind Coronin1c and the latter increases osteoclast differentiation (TRAP assay).

1) In light of the above It should be tested the effects of Rab44 overexpression (WT) on osteoclast differentiation (RANKL-RANK signaling, NFATc1, DC-STAMP, CtsK, MMP9 expression).

Answer: First of all, we appreciate for positive responses for our paper. The suggestion is correct. Actually, our previous study has reported the effects of Rab44 overexpression on osteoclast-differentiation marker genes, including DC-STAMP, OC-STAMP, CTSK, MMP9, RANK, and c-fms. All genes were significantly lower in Rab44-overexpressing osteoclasts compared to control cells (Yamaguchi et al. 2018, Cell Mol Life Sci, 75(1):33-48. Fig. S4).

2) Lanes 77-79 (fig1b.) Four proteins are detected in GFP-Rab44 expressing (WT) cells compared with the GFP-expressing cells (Mock).

2) Is it the expression of Coronin1c protein dependent on the Rab44 overexpression?

Answer: Thank you for your important comments. It depends. However, the active form of Rab44 appears to have decreased expression levels and the inactive form seems to increase. Since Wt is a mixture of active and inactive forms, its effect seems to be weakened.

3) Lanes 88-92. The DN mutant of the Rab domain (T551N) used in this study has not been characterized before (see Tomoko Kadowaki et al. 2021). Is it the short or long form? As these forms are different in that the short one lacks the dominance that binds Calcium.

Answer: Thank you for your useful comment. The DN mutant of the Rab domain (T551N) used in this study is the short form. Therefore, the short form lacks EF-hand motif working as a calcium binding site.

4) Lanes 93-96 (Fig. 1d). The highest protein level of endogenous Coronin1C was observed in DN mutant-expressing cells, although the expression level of Rab44 in the DN mutant-expressing cells was lower than those in WT and the CA mutant expressing cells

4) How could the authors explain this large difference in Rab44 expression level between mutant DN and WT or CA?

Answer: This comment is quite right. The amount of Coronin1C depends on the amount of the GDP form of Rab44. Therefore, it is speculated that Coronin1C and DN are the most stable proteins and the amount is large. On the contrary, since Coronin1C and CA do not bind, the expression level decreases.

5) Lanes 96-98 (Fig 1d). Immunoprecipitation with an anti-GFP antibody revealed that the interaction of Coronin1C and GFP-Rab44 proteins was observed in WT and DN mutant-expressing cells, but not in the mock and CA-expressing cells.

5) How could the authors explain that in mock there is no interaction between Rab44 and Coronin1c?

Answer: Thank you for your good comment. Since the endogenous Rab44 expressed in Mock does not contain GFP, it cannot be detected by immunoprecipitation with GFP antibody. However, you can detect interaction between Rab44 and Coronin1C in the mock using Rab44 antibody.

6) (lanes 98-100). The level of the DN mutant band was higher than that of the WT, suggesting that Coronin1C exclusively interacts with the GDP form of Rab44

6) The Coronic1c protein expression level in the DN mutant was also higher than that of the WT in the lysate (as in IP-GFP). The authors should show a quantization of the Coronin1c protein level also in the lysates.

Answer: According to reviewer’s suggestion, we added the quantitative data of western blotting as shown in Fig. 1e.

7) Lanes 121-126 (figs 2a and 2b). Quantitative analysis shows that Coronin1c binds with Rab44 WT and DN-mutant, and the highest value is in WT cells, differently from what reported in fig. 1d. How do the authors explain this?

Answer: Thank you for your important comments. The suggestion is correct. The PLA method detects protein complex coexistence in a cell. However, we think that this method is qualitative rather than quantitative. Therefore, the data between Fig 1b and Fig 2b seems to be discrepancy.

8) The authors measured Coronin1C expression levels in RAWD cells with or without RANKL addition during 0-3 days (Figure 3a) and found no significant differences. Then, they studied the expression level of the Coronin1c protein (fig 3b) and mRNA (fig 3c) after 3 days of stimulation with RANKL in Mock, WT, DN and CA

8) However, looking at the Mock sample with or without RANKL in Figure 3b, it appears that the expression level of the Coronin1c protein increases. To find differences in osteoclastogenesis, authors should compare the expression levels of the Coronin1c protein and mRNA (WB densitometry, qPCR) by analyzing the same sample / treatment in the presence / absence of RANKL (i.e., Mock-RANKL with Mock + RANKL, WT-RANKL with WT + RANKL, etc.). This way there are differences in the Mock and mutant DN samples.

Answer: This comment is also quite right. According to reviewer’s suggestion, we have changed the arrangement of the quantitative data of Fig. 3b. Also, we added the data of the absence of RANKL (-) of Mock, WT, CA and DN in Fig. 3c.

9) How do the authors explain the big difference of Coronin1c protein expression level in Mock and DN samples or in the WT and DN samples (Figure 3b)?

Answer: Thank you for the good comment. The amount of Coronin1C depends on the amount of the GDP form of Rab44. Therefore, it is speculated that Coronin1C and DN are the most stable proteins and the amount is large. On the contrary, since Coronin1C and CA do not bind, the expression level decreases.

10) In figure 3d the upper and lower panels don't look as the same magnification (Mock-CA and Mock-DN).

Answer: This comment is quite right. It may look like different magnification in Fig. 3d at first glance. I think this is because the size of polynuclear cells is different. However, if you compare the cells of mononuclear cells, you can see that they are the same. Therefore, we have changed the photos in Fig. 3d.

11) In figure 3d the authors analyzed osteoclast formation by comparing CA with Mock and DN with Mock. However, in Figure 1d they demonstrated binding of Coronin1c to Rab44 in WT and DN samples, but not in Mock and CA (-RANKL). Hence, the authors should analyze osteoclast formation in CA and DN-mutant versus overexpressed Rab 44 (WT).

Answer: Thank you for the good comment. According to reviewer’s suggestion, we changed to comparison with Wt instead of comparison with Mock in Fig. 3d

12) Lanes 163-165 (Fig. 3d and 3e). In contrast, the Rab44 DN-expressing osteoclasts showed remarkably larger multinucleated formation than the Mock cells

12) Why in this study, the Rab44 DN-expressing osteoclasts are larger multinucleated whereas in the study of 2018 overexpression of GFP-Rab44 impaired osteoclastogenesis? In the figure 2b the the Rab44 WT-expressing sample is the best binding Coronin1c, why the authors did not use this as control instead Mock?

Answer: The DN mutants of Rab proteins often produce the opposite phenotype to the wild-type overexpressed cells. In other words, the DN mutants of Rab proteins have the same effect as Rab44 knockdown. Indeed, Rab44 knockdown causes larger osteoclast formation compared with the control. According to reviewer’s opinion, instead of using Mock as a control, we remaked the Fig. 2b into a diagram that uses WT as a control.

13) Compared to the control (mock) siRNA, #1, #2, and #3 diminished CORO1C mRNA expression levels by approximately 50%, 15%, and 30%, respectively (Fig. 4a). Coronin1C protein levels were reduced by approximately 60, 60, and 30%, respectively (Fig. 4b). (lanes 193-196)

13) Lanes 193-196. Compared to the control (mock) siRNA, #1, #2, and #3 diminished CORO1C mRNA expression levels by approximately 50%, 15%, and 30%, respectively (Fig. 4a). Coronin1C protein levels were reduced by approximately 60, 60, and 30%, respectively (Fig. 4b).

13) I believe that if you want to indicate in the text that the expression levels diminished by approximately 50, 15 and 30% respectively, in the figure (fig 4a) you must indicate the residuals of the expression level (50, 85 and 70%), or you must write in the text that the residual levels of expression are 50, 15 and 30%. The same for protein expression (Figure 4b).

Answer: Thank you for your important comments. We changed the indication of the residuals of the expression level (50, 15 and 30% in Fig. 4a) and (60, 60 and 30% in Fig. 4b).

14) Why don't the authors knockdown Coronin1c in RAWD cells GFP-Mock, WT, CA and DN Rab44 expressing cells to see the combined effects of the overexpression of Rab44 active and inactive form with the binding to the Coronin1c protein?

Answer: Thank you for your suggestions. According to reviewer’s suggestion, we performed overexpression of Rab44 and knockdown of Coronin1C in RAW-D cells. The data is shown in Fig. 5. The previous Fig. 5, 6, and 7 are moved to Fig. 6. 7. 8, respectively.

Reviewer 2 Report

The manucsript by Yamaguchi et al. describes a new role of Coronin1C in controlling osteoclast formation through interactions with Rab44. The manuscript is well executed and provides new insight into the role of Coronin1C in controlling macrophage motility during osteoclastogenesis. While the manuscript is publishable in its current form, some additional experiments may strengthen the conclusions.

1. The authors rely exclusively on osteoclast counting/size measurements. While this is a valid outcome, additional experimental outcomes would provide greater insight into the effects of Coronin1C on osteoclast formation and activity. Specifically, including the expression of key osteoclast formation and activity genes would greatly complement the formation assays and provide new insight into the impact of Coronin1C on osteoclast biology (e.g., Ctsk, DC-STAMP, OC-STAMP, vATPaseD2).

2. Considering that Coronin1C is an actin-binding protein it would also be advisable to perform F-actin staining to determine if cytoskeleton arrangement is impacted.

Author Response

Comments and Suggestions for Authors

The manucsript by Yamaguchi et al. describes a new role of Coronin1C in controlling osteoclast formation through interactions with Rab44. The manuscript is well executed and provides new insight into the role of Coronin1C in controlling macrophage motility during osteoclastogenesis. While the manuscript is publishable in its current form, some additional experiments may strengthen the conclusions.

  1. The authors rely exclusively on osteoclast counting/size measurements. While this is a valid outcome, additional experimental outcomes would provide greater insight into the effects of Coronin1C on osteoclast formation and activity. Specifically, including the expression of key osteoclast formation and activity genes would greatly complement the formation assays and provide new insight into the impact of Coronin1C on osteoclast biology (e.g., Ctsk, DC-STAMP, OC-STAMP, vATPaseD2).

Answer: First of all, we appreciate for positive responses for our paper. According to reviewer’s suggestion, we performed the quantitative data of osteoclast marker genes, such as DC-STAMP, OC-STAMP, CTSK, and Src, between the control and Coronin 1C-knockdown cells shown in Figure 4e.

  1. Considering that Coronin1C is an actin-binding protein it would also be advisable to perform F-actin staining to determine if cytoskeleton arrangement is impacted.

Answer: This comment is quite important. We added the data of phalloidin staining, which is detectable for F-actin, between the control and Coronin 1C-knockdown cells shown in Figure 4f.

Round 2

Reviewer 1 Report

The manuscript is accepted in the present form